# MITIGATING UNI-MODAL SENSORY BIAS IN MULTIMODAL OBJECT DETECTION WITH COUNTERFACTUAL INTERVENTION AND CAUSAL MODE MULTIPLEXING

## ABSTRACT

Multimodal object detection using RGB and thermal sensors (RGBT) has emerged as a promising solution for safety-critical vision applications that require non-stop operations all day/night. However, there are unsolved issues in multimodal object detection, including uni-modal sensory bias, where models tend to rely on one modality over the other instead of referring to multimodal reasoning. We analyze that training differential multimodal data (i.e., RXTO[1]) on correlation-based symmetrical fusion topology structures provoke such skewed preference. To address this problem, we propose a novel Causal Mode Multiplexing (CMM) framework using the tools of counterfactual intervention. Different from the symmetrical fusion topology of existing methods, the proposed approach leverages two distinct causal graphs based on the input data type. The counterfactual intervention is performed on differential inputs (RXTO, ROTX), while the total effect of the symmetrical fusion topology is learned for common inputs (ROTO). Then, we propose a Causal Mode Multiplexing (CMM) Loss to optimize the interchange between two causal graphs. Overall, the CMM framework enables learning the causality links between the multimodal inputs and predictions, eliminating the uni-modal sensory bias. To assess the effectiveness of CMM, we introduce the ROTX Multispectral Pedestrian (ROTX-MPed) dataset which we will release in public. This dataset mainly includes counterexamples that are not present in existing data. Extensive experiments on KAIST, CVC-14, FLIR, and our ROTX-Mped dataset demonstrate that our CMM framework effectively learns multimodal reasoning and generalizes well on ROTX test data with only training conventional ROTO and RXTO data.

## 1  INTRODUCTION

Object detection is a foundational computer vision task that has achieved significant progress due to the advancements of Deep Neural Networks (DNNs) (Ren et al., 2015; Tan et al., 2020; Redmon & Farhadi, 2017; Hosang et al., 2017; Carion et al., 2020). Such detection models play a critical role in many safety-critical systems such as smart surveillance cameras (CCTVs) and autonomous vehicles (AVs). As these real-world applications are demanded to operate both day and night, technologies leveraging multi-sensory fusion are being supported to develop multimodal object detection. To this end, multimodal object detection utilizing RGB and thermal sensors (RGBT) has evidenced notable progress as the latter can provide solid object signatures under low light and adverse conditions.

Such progress on multimodal object detection is anchored on the infrastructure of multimodal datasets  (Hwang et al., 2015; Choi et al., 2018; FLIR Systems, 2021; González et al., 2016), advanced neural network architectures (Ren et al., 2015; Tan et al., 2020; Redmon & Farhadi, 2017), enhanced multi-sensor fusion technologies  (Kim et al., 2021; Cao et al., 2021), and meticulously devised training methods (Qingyun et al., 2022; Zhou et al., 2020). Despite comprehensive endeavors, there exist several unexplored challenges persist in multimodal object detection. First, one often overlooked factor is the poor generalizability due to *uni-modal sensory bias*. This bias stems when a significant portion of the training data contains scenes where one sensor significantly outperforms

---

[1]R⋆T⋆ refers to the multimodal data type, where ⋆ denotes the perceptibility (O/X) of the ground truth object from the RGB (R) sensor and thermal (T) sensor.

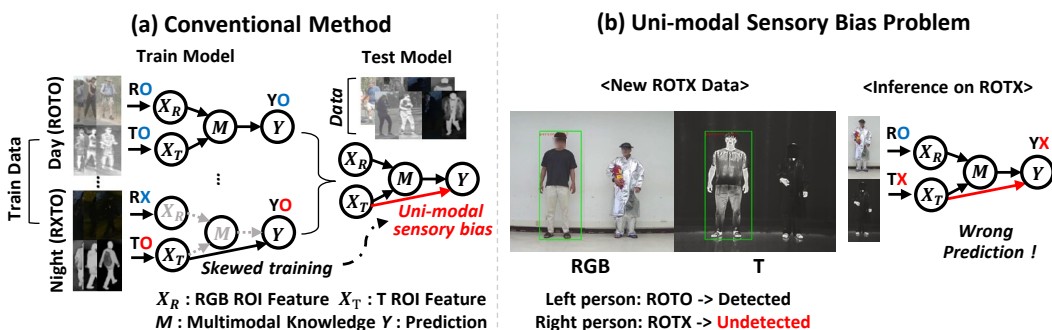

Figure 1: (a) Existing multimodal object detection methods based on a symmetrical fusion topology graph derive uni-modal sensory bias when trained on nighttime (RXTO) data. (b) Left: We discover a new multimodal pair, ROTX data, which is not presented in existing multimodal datasets. Right: Existing models generalize poorly on ROTX data due to uni-modal sensory bias.

the other sensory type. In such scenarios, spurious correlations (i.e., statistical biases) can easily form between the significant modality and target labels, leading to a skewed training preference for one modality in canonical supervised learning. Given that thermal modality plays a significant role in RGBT data, especially in nighttime (RXTO) scenes, models are prone to memorize the strong uni-modal priors in the training set. As a result, models tend to rely on the uni-modal sensory bias as a shortcut instead of multimodal reasoning and thus fail to generalize on counterexamples (e.g., ROTX) as in Fig. 1 (b).

We point out another problem regarding symmetrical fusion topology and correlation-based learning schemes but ignored by previous models. They adapt to the situation of ROTO and guide the model to fuse multimodality with a symmetrical topology graph. However, these methods cannot acquire correct multimodal knowledge when trained on disparate multimodal pairs (RXTO), often prompting the learning of spurious correlations to achieve high accuracy. We argue that uni-modal sensory biases are derived from these learning strategies which often lack of understanding causal relations between the multimodal inputs and predictions.

In this paper, we propose a novel Causal Mode Multiplexing (CMM) framework to mitigate uni-modal sensory bias and guide correct multimodal reasoning based on causality. Specifically, the CMM framework facilitates the multimodal learning process in 1) common mode, and 2) differential mode causal graphs. First, common mode is defined as the situation where the model gets common inputs, such as daytime (ROTO) data. For the common mode, we guide the model to learn the *total effect* (Pearl, 2022) of the symmetrical fusion topology structure. Second, the differential mode is when differential inputs (RXTO, ROTX) are given. In this case, we utilize the tools of counterfactual intervention to eliminate the effect of the uni-modal sensory bias. We formulate uni-modal sensory bias as the direct link and eliminate the bias by subtracting it from the total effect. To this end, we modify the training objective from maximizing the posterior probability likelihood to maximizing the *total indirect effect* (Pearl, 2022). Finally, to combine two different learning schemes and guide the CMM framework to optimize the interchange between two causal graphs, we propose a Causal Mode Multiplexing (CMM) Loss.

Furthermore, we discover intriguing ROTX samples, unobserved in the existing datasets (Hwang et al., 2015; González et al., 2016; FLIR Systems, 2021), that challenge models due to their inherent uni-modal sensory biases. In most of the existing datasets, instances are either ROTO (day) or RXTO (night). Different from these data, ROTX samples are only perceivable in RGB which causes models to produce false-negative errors attributed to their unimodal bias as showcased in Fig.1 (b). To test the generalizability of the CMM framework, we collect a new ROTX multispectral pedestrian dataset, namely ROTX-MPed. ROTX-MPed comprises 1.5k RGBT image pairs captured from real-world scenes highly related to critical applications of multimodal object detection. Our experimental results demonstrate that our CMM framework generalizes well under ROTX-MPed even without ROTX samples in the training set while performing robustly on the conventional datasets.

The main contributions of our paper are:

1. To the best of our knowledge, this is the first work exploring uni-modal sensory bias in a multisensor fusion model. Especially, we introduce the ROTX Multispectral Pedestrian (ROTX-MPed) dataset, which presents a notable challenge to multimodal object detection models due to their inherent uni-modal sensory biases.

2. We propose a Causal Mode Multiplexing (CMM) framework that interchangeably learns multimodal representations between two distinct causal graphs. Consequently, our CMM framework learns the causality links between the multimodal inputs and predictions.

3. Extensive experiments demonstrate that our CMM framework generalizes well under ROTX test data with even ROTO and RXTO training data.

## 2 RELATED WORK

### 2.1 MULTISPECTRAL PEDESTRIAN DETECTION

Multimodal object detection jointly leverages multiple sensors to help machines locate target objects better. Using RGBT in particular, becomes attractive in applications such as autonomous vehicles (AVs) that face low-light or adverse environments. Current research on multimodal object detection mainly focuses on developing effective fusion strategies. For instance, MBnet (Zhou et al., 2020) adaptively fuses the RGBT complementary features according to illumination conditions. Recent works improve each modal feature through cross-modal learning. Kim et al. (2022) proposed an uncertainty-aware feature fusion (UFF) network that alleviates miscalibration and modality discrepancy problems. Cross-Modality Fusion Transformer (Qingyun et al., 2022) introduces a new cross-modality fusion mechanism based on self-attention. However, they all suffer from the uni-modal sensory bias problem, constraining practical applicability. In this paper, we propose a novel CMM framework to mitigate the bias by learning multimodal causal representations, achieving satisfying generalization.

### 2.2 CAUSALTY-INSPIRED MACHINE LEARNING

Causal inference and counterfactual reasoning encourage machines to explore causality behind observational likelihood, a proven and effective analytical approach in many machine learning problems (Monteiro et al., 2022; Wu et al., 2021). Several works leveraged counterfactual reasoning aiming to endow models with the capability to explore and understand causal effects. Niu et al. (2021) mitigate the direct language effect on visual question answering (VQA) by guiding the model to learn the total indirect effect (TIE). Zhang et al. (2023) introduce the layout-based soft Total Direct Effect (L-sTDE) to adjust the prediction of the navigation policy in object navigation. Different from them, we propose a Causal Mode Multiplexing (CMM) framework which interchangeably learns between total effect (TE) and total indirect effect (TIE) depending on the data type.

## 3 PRELIMINARIES

### 3.1 STRUCTURAL CAUSAL MODEL (SCM)

Before introducing our method, we present the fundamental concepts of causal inference (Pearl et al., 2000; Pearl, 2022; Pearl & Mackenzie, 2018; Robins, 2003).
Structural Causal Models reflect the cause-and-effect relationships (*links* $\mathcal{E}$) between the set of variables (*nodes* $\mathcal{V}$). The causality links (*cause→effect*) are represented in a acyclic graph $\mathcal{G} = \{\mathcal{V}, \mathcal{E}\}$. For random variables $X$ and $Y$ which the direct effect of $X$ is on $Y$, the causality link could be formulated as $X \rightarrow Y$. If an indirect effect of $X$ is on $Y$ through the variable $M$, $M$ is considered a mediator between $X$ and $Y$ ($X \rightarrow M \rightarrow Y$ in Fig.2 (a)). With structural causal models, the examination of causality links among variables can be achieved through variable intervention, which involves modifying the value of particular variables and subsequently observing the outcomes.

### 3.2 COUNTERFACTUAL INTERVENTION

Counterfactual intervention can break the causality link and eliminate the effect of particular variables. The key point of counterfactual intervention is to measure the effect $Y$ from two different treatments to the cause variable $X$: factual and counterfactual. Take the SCM in Fig. 2 (a) as an example i.e., $X \rightarrow M \rightarrow Y$. Suppose that $X = x$ represents the "treatment condition" and $X = x^\star$

represents the "no-treatment condition" (lowercase letter indicates the observed value of the random variable). Then we can consider factual and counterfactual scenarios.

**Notations**. We denote the value that $Y$ would obtain in the factual scenario if $X$ is assigned $x$ and $M$ assigned $m$ as $Y_{x,m_x} = Y(X = x, M = m_x)$. Similarly, in the counterfactual scenario, $Y$ would have the value $Y_{x,m_{x^\star}} = Y(X = x, M = m_{x^\star})$ when $X = x$, $M = m_{x^\star} = M(X = x^\star)$. $X$ is set to a value $x^\star$ (usually zero value or mean features) and can break the causality link between $M$ and its parent node $X$. We note that in the counterfactual scenario, $X$ can be simultaneously assigned to different values $x$ and $x^\star$. So when the intervention is conducted on $M$, the variable $X$ retains its original value of $x$ as if $x$ (observation on $X$) had existed as in Fig.2 (b).

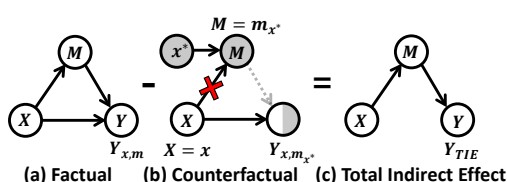

Figure 2: Strcture Causal Models (SCMs) of (a) factual, (b) counterfactual, and (c) total indirect effect scenarios. The direct effect of $X \to Y$ can be eliminated due to counterfactual intervention.

**Total Indirect Effect**. Now we can estimate the total indirect effect (TIE) by comparing two hypothetical scenarios.

$$TIE = Y_{x,m_x} - Y_{x,m_{x^\star}}. \tag{1}$$

Total indirect effect (TIE), as in Fig.2 (c) breaks the causality link of $X \to Y$ and eliminates the direct effect $Y$ of cause $X$. Furthermore, Total indirect effect (TIE) can be decomposed into total effect (TE) and natural direct effect (NDE).

$$
\begin{aligned}
TIE &= TE - NDE \\
&= (Y_{x,m_x} - Y_{x^\star,m_{x^\star}}) - (Y_{x,m_{x^\star}} - Y_{x^\star,m_{x^\star}}).
\end{aligned} \tag{2}
$$

When we examine the total effect (TE), we compare two hypothetical scenarios: one where $X = x$ and the other where $X = x^\star$. In contrast, the natural direct effect (NDE) represents the effect of $X$ on $Y$ while keeping the mediator $M$ blocked. It measures the change in $Y$ as $X$ transitions from $x^\star$ to $x$, with $M$ assigned to the value of the no-treatment $X = x^\star$, thereby nullifying $M$'s response to the treatment $X = x$. In the subsequent section, we will take a more in-depth look at the interpretations of these effects in the context of multimodal object detection.

## 4 STRUCTURAL CAUSAL MODEL OF MULTIMODAL OBJECT DETECTION

Before performing counterfactual interventions, we first formulate the task in a causal graph. Multimodal object detection models require RGB and thermal input images and base features are extracted using uni-modal encoders. Then RGB and thermal ROI (Region of interests) features, denoted as $X_R = x_R$ and $X_T = x_T$, are generated. These features are fused, producing fusion features of multimodal knowledge $M$ that contain the information of both RGB and thermal. Final predictions are made through the head network and predict class labels $Y$. Bounding boxes are determined by Non-maximum suppression (NMS).

Now we formulate this multimodal object detection framework in a structural causal model (SCM). We delve into the causal relationship between the multimodal knowledge $M$ and the predicted class labels $Y$, which differs for daytime (ROTO) and nighttime (RXTO) data. For daytime (ROTO) data, the class label $Y$ can be derived directly from the multimodal knowledge $M$ which contains common informations of both modalities. On the other hand, for nighttime (RXTO) data, the multimodal knowledge $M$ contains differential information about each modality. In this case, the intended behavior of the model is to make predictions based on correct reasoning based on the multimodal knowledge $M$. However, machine-learning models learn simple correlations based on maximizing likelihoods, as class labels $Y$ are skewed toward thermal $x_T$. We formulate this uni-modal sensory bias as the direct link between $X_T$ and $Y$ ($X_T \to Y$). Bringing such factors at once, we can represent two causal graphs for each ROTO and RXTO scenario as shown in Fig.3 (a)-(b). The below describe each node and causality link in detail.

**Link $X_R \rightarrow M \leftarrow X_T$ (Feature Fusion).** RGB and thermal ROI features ($X_R$ and $X_T$) are fused to generate ROI fusion features of multimodal knowledge $M$.

**Link $M \rightarrow Y$ (Class Label Prediction with Multimodal Knowledge).** Using the multimodal knowledge $M$, the head network outputs class prediction scores $Y$ for each fused ROI feature.

**Link $X_T \rightarrow Y$ (Uni-modal Sensory Bias)** This direct link is unintentionally formed during training on nighttime (RXTO) data. We aim to eliminate this direct link and guide the model to correctly utilize multimodal knowledge $M$ for predicting $Y$.

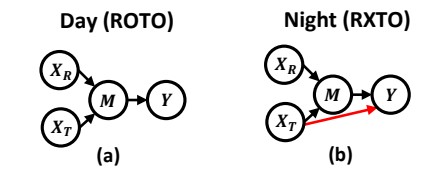

Figure 3: Strcture Causal Models (SCMs) on training (a) ROTO and (b) RXTO samples. (b): The uni-modal sensory bias can be viewed as the direct effect on $Y$ of $X_T$.

## 5 PROPOSED METHOD

### 5.1 CAUSAL MODE MULTIPLEXING

**Learning Two Distinct Causal Graphs.** Our proposed method is strategically designed to interchangeably learn between *common* and *differential* modes based on the following rationales: 1) First, *common mode* is when ROTO input is given. We learn the causal graph devised by the symmetrical fusion topology, as it can effectively guide correct multimodal reasoning on ROTO. 2) Second, the differential mode is when differential (RXTO, ROTX) inputs are given. In such cases, the model is guided to learn the "total indirect effect" which is based on a counterfactual intervention to prune the direct effect of the uni-modal sensory bias. Note that our purpose is to learn multimodal causality from ROTO and RXTO contained in datasets to obtain satisfactory generalizability on ROTX. Thus we use ROTO and RXTO samples and not ROTX when training.

**1. Common Mode - ROTO Train/Test Graph.** We formulate the structural causal model of *common mode* as in Fig.4 (a). Also, we intentionally add the link $X_R \rightarrow Y$ and $X_T \rightarrow Y$ to estimate the uni-modal direct effect of $X_R$ on $Y$ and $X_T$ on $Y$. Adding these links provides a way to measure direct effects while preserving the symmetrical fusion topology. To implement this direct link, we train a uni-modal neural model denoted as $H_{\theta_{X_R}}(\cdot)$ and $H_{\theta_{X_T}}(\cdot)$. We denote the prediction scores obtained from direct links as $Y_{x_R}^{CM}$ and $Y_{x_T}^{CM}$ which can be expressed as below:

$$Y_m^{CM} = H_{\theta_M}(m), Y_{x_R}^{CM} = H_{\theta_{X_R}}(x_R), Y_{x_T}^{CM} = H_{\theta_{X_T}}(x_T), \tag{3}$$

where the neural network $H_{\theta_M}(\cdot)$ refers to the head network which takes fusion ROI features $M = m$ and outputs the class scores $Y_m^{CM}$ (direct link $M \rightarrow Y$).
Then, we design a fusion function $\mathcal{F}(\cdot)$ with respect to $Y_m^{CM}$, $Y_{x_R}^{CM}$, and $Y_{x_T}^{CM}$ i.e., $\mathcal{F}(Y_m, Y_{x_R}, Y_{x_T})$ to produce a single prediction score $Y_{m,x_R,x_T}^{CM}$. From general fusion methods, we use the nonlinear Log-Harmonic (LH):

$$Y_{m,x_R,x_T}^{CM} = Y_{HM}^{CM} = log(\sigma(Y_m^{CM}) \times \sigma(Y_{x_R}^{CM}) \times \sigma(Y_{x_T}^{CM})), \tag{4}$$

where $\sigma$ denotes the sigmoid function.

**Total Effect (TE).** For the common mode, we measure the total effect (TE). Note that total effect considers the "factual scenario" where $X_R$, $X_T$, and $M$ are all accessible. Following the definition from the preliminary section, the total effect of the common mode causal graph can be written as:

$$TE = Y_{m,x_R,x_T}^{CM} - Y_{m^\star,x_R^\star,x_T^\star}^{CM}. \tag{5}$$

where $Y_{m^\star,x_R^\star,x_T^\star}^{CM}$ is the no-treatment condition. The implementation details of the no-treatment condition are described in the *supplementary material*.

**2. Differential Mode - RXTO Train/RXTO, ROTX Test Graph.** Different from the common mode, we intentionally assign node values and prune the causal link $X_T \rightarrow Y$. To this end, we introduce the natural direct effect.

**Natural Direct Effect (NDE).** We estimate the natural direct effect (NDE), the "counterfactual scenario", which refers explicitly to the uni-modal sensory bias i.e., $X_T \rightarrow Y$ in Fig.4 (b). To

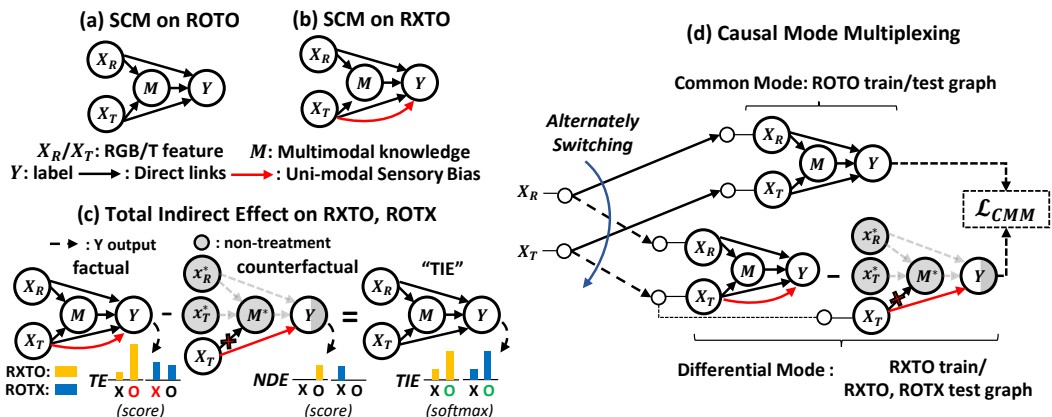

Figure 4: Our Structural Causal Model (SCM) formulations on (a) ROTO (day) and (b) RXTO (night) data. (a) We add links ($X_R \to Y$, $X_T \to Y$) to the conventional causal graph to measure uni-modal direct effects. (b): The uni-modal sensory bias (red) hinders multimodal reasoning. (c) implies the total indirect effect on RXTO and ROTX which the direct effect of uni-modal sensory bias is pruned. (d) We propose a Causal Mode Multiplexing framework that learns correct multimodal reasoning both on ROTO and RXTO data.

achieve this goal, we measure the direct effect of $X_T = x_T$ on $Y = y$ by blocking the effect of $X_R$ and $M$. Using the no-treatment condition definitions in the preliminary section, $X_T$ is set to $x_T$ and $M$ would attain the value $m^\star$ when $X_T$ had been $x_T{}^\star$ and $X_R$ is $x_R{}^\star$. We obtain the natural direct effect (NDE) of $X_T$ on $Y$ by comparing the counterfactual graph to the no-treatment conditions:

$$NDE = Y^d_{m^\star, x_R{}^\star, x_T} - Y^d_{m^\star, x_R{}^\star, x_T{}^\star},\tag{6}$$

where $m^\star$ and $x_T{}^\star$ represent the no-treatment condition.

**Total Indirect Effect (TIE).** The elimination of uni-modal sensory bias can be achieved by subtracting the natural direct effect (NDE) from the total effect (TE).

$$TIE = TE - NDE = Y^{CM}_{m, x_R, x_T} - Y^d_{m^\star, x_R{}^\star, x_T}.\tag{7}$$

Fig. 4 (c) shows the illustration of the total indirect effect of the differential mode. Note that the no-treatment conditions $Y^d_{m^\star, x_R{}^\star, x_T{}^\star}$ and $Y^{CM}_{m^\star, x_R{}^\star, x_T{}^\star}$ are the same. For inference, we opt for the label with the highest TIE, in contrast to the conventional strategies that primarily rely on posterior probability i.e., $p(y|x_R, x_T)$.

## 5.2 CAUSAL MODE MULTIPLEXING (CMM) LOSS

**Determination of Causal Modes**. To assign causal modes in Fig.4 (d), we leverage the uni-modal prediction scores $Y^{CM}_{x_R}$ and $Y^{CM}_{x_T}$ based on the following rationale. For common mode (ROTO inputs), both of the $Y^{CM}_{x_R}$ and $Y^{CM}_{x_T}$ value difference between the foreground and the background will have a same sign while an opposite sign will happen if differential inputs are given such as RXTO or ROTX. Using these properties, we design a binary translation function to calculate the mode number. We denote the mode number denoted as $K_{mode}$, for which we want the value 1 for the common mode and -1 for the differential mode. The details of our function design are as follows. Let $\pi_R = [\pi_R^b, \pi_R^f]$ and $\pi_T = [\pi_T^b, \pi_T^f]$ the approximate representation of the one-hot vector of the prediction labels derived from the $argmax$ of $Y^{CM}_{x_R}$ and $Y^{CM}_{x_T}$ where index 0 refers to the prediction label of being the background, and index 1 to the prediction label of being the foreground (object). Since the $argmax$ operation is non-differentiable on the gradient descent method, we adopt the Gumbel-softmax (Jang et al., 2016) estimation. Then we can write $\pi_R$ and $\pi_T$ as:

$$\pi_R = \text{softmax}\left[g_R + log(Y^{CM}_{x_R})/\tau\right], \pi_T = \text{softmax}\left[g_T + log(Y^{CM}_{x_T})/\tau\right]\tag{8}$$

where we set the Gumbel noises $g_R$ and $g_T$ to zero since we do not need random sampling variations for our purpose. From the formula of eq.(8), we can obtain values $\left\{\pi_R^b = 0, \pi_R^f = 1, \pi_T^b = 0, \pi_T^f = 1\right\}$ for ROTO, and $\left\{\pi_R^b = 1, \pi_R^f = 0, \pi_T^b = 0, \pi_T^f = 1\right\}$ for

RXTO, and $\left\{ \pi_R^b = 0, \pi_R^f = 1, \pi_T^b = 1, \pi_T^f = 0 \right\}$ for ROTX. From these values, we can determine the causal mode number $K_{mode}$ according to the above rationale.

$$K_{mode} = \Delta\pi_R \times \Delta\pi_T = (\pi_R^f - \pi_R^b) \times (\pi_T^f - \pi_T^b). \tag{9}$$

This $K_{mode}$ value will obtain 1 for the common mode and -1 for the differential mode. Then we can design a "switchable total indirect effect" (sTIE) in which the cause-effect is differently calculated between TE and TIE according to the causal mode number $K_{mode}$.

$$sTIE = TE - ReLU(-K_{mode}) \times NDE = Y_{m,x_R,x_T}^{CM} - ReLU(-K_{mode}) \times Y_{m^\star,x_R^\star,x_T}^d$$

$$= \begin{cases} Y_{m,x_R,x_T}^{CM} & if \ K_{mode} = 1 \ \textit{(Common mode)} \\ Y_{m,x_R,x_T}^{CM} - Y_{m^\star,x_R^\star,x_T}^d & if \ K_{mode} = -1 \ \textit{(Differential mode)}. \end{cases} \tag{10}$$

in which sTIE is calculated as the total effect in eq.(5) for ROTO inputs and as the total indirect effect in eq.(7) for RXTO and ROTX inputs.

**Causal Mode Multiplexing Loss**. We formulate the sTIE in a loss function. Before introducing our Causal Mode Multiplexing Loss, we first revisit the loss function of the conventional model. Given a triplet $(x_T, x_R, y)$ where $y$ is the ground-truth class label of RGB/T ROI pair:$x_R/x_T$, the ROI classification branches of the conventional multimodal object detection model (Kim et al., 2022) is optimized by:

Table 1: Cause-effect and $K_{mode}$ assignment with respect to data type.

| Type | $\Delta\pi_R$ | $\Delta\pi_T$ | $K_{mode}$ | |
|------|------|------|------|------|
| ROTO | 1 | 1 | 1 | TE |
| RXTO | -1 | 1 | -1 | TIE |
| ROTX | 1 | -1 | -1 | TIE |

$$\mathcal{L}_{cls} = \mathcal{L}_Y(TE, y) \ \textit{(Conventional)}. \tag{11}$$

Where L denotes the cross-entropy loss. This conventional loss guides the model to learn the total effect on both ROTO and RXTO, which provoke uni-modal sensory bias toward thermal. Different from them, Causal Mode Multiplexing (CMM) loss learns the correct multimodal reasoning for both ROTO and RXTO training data based on two causal modes:

$$\mathcal{L}_{CMM} = \mathcal{L}_Y(sTIE, y), \tag{12}$$

which sTIE refers to the formula in eq.(10). The overall classification branch loss can be written as:

$$\mathcal{L}_{cls} = \mathcal{L}_{CMM} + \mathcal{L}_Y(x_R, y) + \mathcal{L}_Y(x_T, y) \ \textit{(Ours)}, \tag{13}$$

where $\mathcal{L}_Y(x_R, y)$ and $\mathcal{L}_Y(x_T, y)$ are over $Y_{x_R}^{CM}$ and $Y_{x_T}^{CM}$.

## 5.3 IMPLEMENTATION.

**Training**. The final training loss is the combination of $\mathcal{L}_{cls}$, bounding box regression loss $\mathcal{L}_{bbox}$, and $\mathcal{L}_{model}$ which include the RPN and the uncertainty module. We follow the implementation details of the paper (Kim et al., 2022).

$$\mathcal{L}_{total} = \sum_{(x_R,x_T,y)\in D} \mathcal{L}_{cls} + \mathcal{L}_{bbox} + \mathcal{L}_{model}. \tag{14}$$

**Inference**. We use the switchable total indirect effect (sTIE) causal effect for inference.

$$sTIE = Y_{m,x_R,x_T}^{CM} - ReLU(-K_{mode}) \times Y_{m^\star,x_R^\star,x_T}^d. \tag{15}$$

## 6 NEW ROTX DATASET

Since conventional datasets (Hwang et al., 2015; FLIR Systems, 2021) rarely contain ROTX samples, we construct a new dataset, namely: the ROTX Multispectral Pedestrian (ROTX-Mped) Dataset. ROTX-Mped is comprised of 1500 RGBT pairs (train: 500 / test: 1000) in two practical scenarios related to the critical applications of multimodal object detection.

**1. Pedestrian Over a Glass Window.** Multimodal object detection is attractive for all-day/night smart surveillance. However, pedestrians observed through a window are perceivable in RGB but invisible in thermal, because thermal radiation can not penetrate through glass. The uni-modal sensory bias restricts then multimodal detection range within the window.

**2. People Wearing Heat-insulation Cloths.** Heat-insulation clothing such as fire protection gears or low-emissivity clothing provides a way of thermal invisibility or *stealth*. Wearing fire protection gear can make firefighters or evacuees undetectable from autopilots in rescue situations. Also, criminals (e.g., bank robbers) can wear low-emissivity clothing to evade multimodal security cameras. We will release the collected dataset in public for future research.

Table 2: **Comparison on ROTX-Mped and conventional (KAIST, CVC-14, and FLIR) test sets** (without extra ROTX training samples). **Best** and **runners up** results obtained are highlighted.

| Train | | KAIST | | | | CVC-14 | | | | FLIR | |
|---|---|---|---|---|---|---|---|---|---|---|---|
| Test | | ROTX-MPed | KAIST | | | ROTX-MPed | CVC-14 | | | ROTX-MPed | FLIR |
| Metric | | AP(↑) | MR(↓) | | | AP(↑) | MR(↓) | | | AP(↑) | AP(↑) |
| | | All | Day | Night | All | All | Day | Night | All | All | All |
| *Uni-modality Network* | | | | | | | | | | | |
| Faster R-CNN | RGB | 60.61 | 27.45 | 42.27 | 32.18 | 34.53 | 96.34 | 82.43 | 93.10 | 55.08 | 34.51 |
| Faster R-CNN | T | 5.33 | 26.70 | 9.73 | 20.79 | 4.66 | 40.14 | 25.97 | 33.81 | 2.06 | 47.03 |
| YOLO-v5 | RGB | **65.5** | 22.98 | 36.08 | 27.33 | 35.1 | 88.96 | 79.64 | 84.22 | 75.7 | 57.3 |
| YOLO-v5 | T | 4.78 | 22.12 | 7.22 | 17.65 | 9.51 | 30.86 | 15.34 | 23.85 | 4.64 | 74.3 |
| *Multi-modality Network* | | | | | | | | | | | |
| Halfway Fusion | RGB+T | - | 24.88 | 26.59 | 25.75 | - | 36.29 | 26.29 | 31.99 | - | - |
| +Faster R-CNN | RGB+T | 36.95 | 14.24 | 10.26 | 13.04 | 8.8 | 50.48 | 36.05 | 44.31 | 13.21 | 71.2 |
| CFT | RGB+T | 3.64 | 14.55 | 8.3 | 12.29 | 8.58 | - | - | - | 5.28 | 84.1 |
| MBNet | RGB+T | 18.88 | **9.64** | 8.26 | 9.04 | - | **24.7** | 13.5 | 21.1 | - | - |
| Kim et al. | RGB+T | 21.69 | 10.11 | **5.05** | **8.67** | 13.36 | **23.87** | **11.08** | **18.70** | 12.23 | **84.6** |
| CMM (Ours) | RGB+T | **70.44** | **9.6** | **5.93** | **8.54** | **34.96** | 27.81 | **7.71** | **17.13** | **57.09** | **87.8** |

## 7 EXPERIMENTS

The CMM framework is introduced to assess the generalizability of multimodal object detection models when there are substantial differences in the RGBT modality-pair distributions between the training and test splits. We mainly test CMM on our collected ROTX-MPed dataset, trained from conventional data splits: KAIST (Hwang et al., 2015), CVC-14 (González et al., 2016), FLIR (FLIR Systems, 2021) train data. Moreover, we report the experimental results on the KAIST, CVC-14, and FLIR test data to check whether CMM over-corrects the unimodal sensory bias. Model performance is evaluated via Log Average Miss-Rate (MR↓) - KAIST, CVC-14, and Average Precision (AP↑) - FLIR, ROTX-Mped. A low MR and high AP value indicate high detection performance. We implement CMM based on the baseline model (Kim et al., 2022) proposed, to interchangeably estimate between total effects and total indirect effects according to eq.(10). Evaluation of our method is compared with five baseline multimodal object detection architectures: Uncertainty-Guided (Kim et al., 2022), Cross-modality Fusion Transformer (CFT) (Qingyun et al., 2022), MB-Net (Zhou et al., 2020), Halfway Fusion (Park et al., 2018), and Halfway Fusion+Faster RCNN version. The experimental settings are described in detail in the supplementary material.

### 7.1 QUANTITATIVE RESULTS

The main results are reported in Table 2. Most of the previous multi-modality methods severely fail on ROTX-Mped test data when trained on conventional datasets. Compared to the RGB uni-modality model (Yolo-v5 (et. al., 2021) and Faster R-CNN (Ren et al., 2015)), the AP of previous multi-modality methods trained on KAIST, CVC-14, and FLIR dataset drops by an average of 42.51, 44.03, and 55.46 AP, respectively. This result supports our original claim that models learn uni-modal sensory bias from conventional data and fail to generalize on ROTX. On the other hand, CMM shows superior generalizability on ROTX-Mped test data compared to other multi-modality models, which indicates that uni-modal sensory bias is effectively reduced. Not only that, the CMM framework achieves great performance on conventional test sets (KAIST, CVC-14, and FLIR) validating that CMM does not over-correct uni-modal sensory bias. Furthermore, we evaluate the effectiveness of CMM compared to existing debiasing strategies via cause-effects. TE+TIE and TIE+TIE train/inference are considered. The results are shown in Table 3. From the results, it can be viewed that TE training is effective in learning conventional data but exhibits poor generalizability on ROTX samples. TIE training enhances model generalizability for ROTX data but degrades on conventional data. Compared to them, the CMM framework is designed to adaptively learn between TE and TIE and show robust performance on both ROTX and conventional test sets. Furthermore, Table 4 attests to the comparable performance of CMM when compared to conventional methods trained with ad-

Table 3: **The comparison between CMM and existing debiasing strategies.** "A+B" denotes the strategies that train the model with "A" cause-effect and test with "B" cause-effect.

| Train | KAIST | | | | Train | CVC-14 | | | | Train | FLIR | |
|---|---|---|---|---|---|---|---|---|---|---|---|---|
| Test | ROTX-Mped (AP ↑) | KAIST (MR ↓) | | | Test | ROTX-Mped (AP ↑) | CVC-14 (MR ↓) | | | Test | ROTX-Mped (AP ↑) | FLIR (AP ↑) |
| | All | Day | Night | All | | All | Day | Night | All | | All | All |
| Baseline | 21.69 | 9.68 | **5.80** | 8.79 | Baseline | 13.36 | **23.87** | 11.12 | 19.21 | Baseline | 12.23 | 84.67 |
| TE+TIE | 57.05 | 26.89 | 27.66 | 27.27 | TE+TIE | 27.97 | 32.85 | 12.11 | 22.33 | TE+TIE | 19.33 | 64.27 |
| TIE+TIE | 56.45 | 12.53 | 9.24 | 12.10 | TIE+TIE | 27.75 | 33.53 | 9.27 | 21.63 | TIE+TIE | 12.02 | 79.39 |
| **CMM** | **70.44** | **9.6** | 5.93 | **8.54** | **CMM** | **34.96** | 27.81 | **7.71** | **17.13** | **CMM** | **57.09** | **87.8** |

Table 4: **Comparison of CMM and a conventional model.** Note that only the conventional model is trained with extra ROTX samples.

| Train | KAIST+ROTX-Mped | | | | Train | CVC-14+ROTX-Mped | | | | Train | FLIR+ROTX-Mped | |
|---|---|---|---|---|---|---|---|---|---|---|---|---|
| Test | ROTX-Mped (AP↑) | KAIST (MR↓) | | | Test | ROTX-Mped (AP↑) | CVC-14 (MR↓) | | | Test | ROTX-Mped (AP↑) | FLIR (AP↑) |
| | All | Day | Night | All | | All | Day | Night | All | | All | All |
| Kim et al. | **72.68** | 30.12 | 29.52 | 29.30 | Kim et al. | **35.27** | 70.30 | 32.19 | 54.16 | Kim et al. | 55.92 | 63.30 |
| **Ours** | 70.44 | **9.6** | **5.93** | **8.54** | **Ours** | 34.96 | **27.81** | **7.71** | **17.13** | **Ours** | **57.09** | **87.8** |

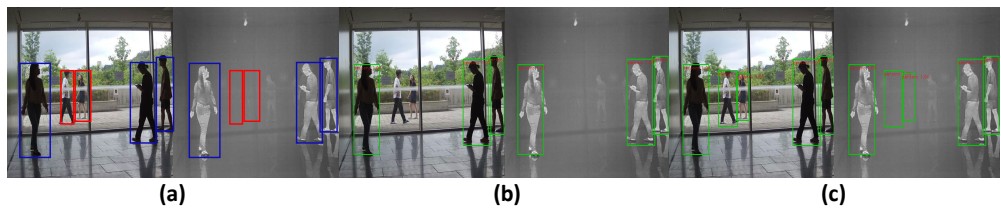

(a)  (b)  (c)

Figure 5: **Qualitative comparison on ROTO (blue), and ROTX (red) test data**. (b): Conventional models fail to detect ROTX pedestrians due to uni-modal sensory bias. (c): Our CMM framework estimates the total effect for ROTO and the total indirect effect for ROTX samples. As a result, it produces correct detection results for ROTX samples.

ditional ROTX training data. As conventional methods improve accuracy on the ROTX-Mped test by finetuning, the performance on KAIST, CVC-14, and FLIR test sets drops severely. Comparably, CMM achieves high performance on both ROTX-MPed and conventional datasets, even without training ROTX samples.

### 7.2 QUALITATIVE RESULTS

The qualitative results in Fig.5 serve as validation for the effectiveness of the CMM framework in mitigating unimodal sensory bias while preserving multimodal context. As depicted in Figure 5, the CMM framework effectively addresses unimodal bias in ROTX samples and produces detection outputs based on correct multimodal knowledge. Specifically, the CMM framework computes the total effect for the ROTO samples marked as blue, and the total indirect effect for the ROTX samples marked as red in Fig.5 (a). In contrast, baseline models fail due to the memorized priors in the training set. Additionally, CMM consistently delivers high-confidence scores and accurate detection boxes.

## 8 CONCLUSION

In this paper, we reveal the uni-modal sensory bias problem in multimodal object detection and present our innovative solution: Causal Mode Multiplexing (CMM) framework. Leveraging the tools of counterfactual intervention, CMM enables the model to interchangeably learn between two distinct causal graphs depending on the input data type. A Causal Mode Multiplexing (CMM) Loss is proposed to optimize the interchange between two causal graphs. Additionally, we introduce the ROTX Multispectral Pedestrian (ROTX-MPed) dataset, which includes counterexamples not included in the conventional data to challenge existing models and test the effectiveness of CMM. Experimental results on KAIST, CVC-14, FLIR, and our ROTX-Mped dataset demonstrate that CMM effectively learns multimodal reasoning and even performs well on ROTX test data with training ROTO and RXTO data.

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

## A  APPENDIX

### A.1  DATA COLLECTION

For data collection purposes, we securely mounted the cameras on tripods and recorded videos capturing 89 surveillance scenarios, including building entrances, hallways, and urban streets. We employed the FLIR Duo Pro R camera, a product by FLIR Systems, Inc., which supports both RGB and IR imaging (with a wavelength range of approximately 7.5-13.5m) simultaneously in a pip dual mode. Twenty volunteers enthusiastically participated in the data collection process, with each individual consenting to both the procedure and the subsequent release of the data to the public. To safeguard privacy, facial features were intentionally blurred. Volunteers were photographed in various indoor/outdoor settings both day/night, with the camera's distance from them varying between 1 to 15 meters. Each video involved 2 to 8 volunteers as actors. From the recorded videos, we meticulously curated a dataset comprising a total of 1500 pairs (500 train/ 1000 test) of visible and thermal images. We took great care to ensure that all visible-thermal image pairs were precisely aligned, maintaining a Field-of-View (FOV) of $32° \times 26°$ and an image resolution of 640×512. High-quality image pairs featuring pedestrians were meticulously synchronized and handpicked during the curation process. We will make the data publicly available after acceptance.

## A.2 Experimental Setting

**Datasets**. In our work, we conducted experiments on three public datasets on multimodal object detection. The first of these datasets is the KAIST Multispectral Pedestrian Detection dataset, which we refer to as the KAIST dataset (Hwang et al., 2015). This dataset comprises an extensive collection of 95,328 RGBT image pairs, featuring 103,128 pedestrian annotations and 1,182 identities. The other one is CVC-14 (González et al., 2016) visible and thermal image pairs. Note that visible images in CVC-14 are in grayscale. For our experiments with the KAIST dataset, we followed the evaluation protocol outlined in (Li et al., 2019; Zhang et al., 2019; Zhou et al., 2020), utilizing the annotation labels provided by the authors of (Hwang et al., 2015). Our evaluation was carried out on a test set consisting of 2,252 images, which we categorized as 'All.' The test set can be divided into 1,455 daytime ('Day') and 797 nighttime ('Night') images. The image resolution for the KAIST dataset was maintained at $512 \times 640$ pixels. Turning to the CVC-14 dataset (González et al., 2016), we leveraged 7,085 train, and 1,433 test images, all with a resolution of $640 \times 471$ pixels. For FLIR, we use the (Zhang et al., 2020) which provides an "aligned" version. Composed of 5,142 aligned images of $640 \times 512$ resolution, divided into 4,129 training data and 1,013 for testing.

**Baseline Model** We constructed our CMM framework based on the Uncertainty-guided model (Kim et al., 2022), with the FPN architecture with ResNet-50 (He et al., 2016) as backbone networks. We follow the implementations in the paper. Specifically, we use the pytorch library and stochastic gradient descent (SGD) for optimization, synchronizing the process across 4 GTX 1080 Ti GPUs. Each GPU handles 2 images, resulting in a total of 8 images per mini-batch. Consistent with (Kim et al., 2022), we first train the model 3 epochs. The learning rate is initialized at 0.006 during the initial 2 epochs and subsequently applied a 0.1 learning rate decay for the final epoch. We configured the number of Region of Interests (RoIs) per image to N=300. For other baseline models that we compare with, such as (Zhou et al., 2020; Qingyun et al., 2022) we use the code and the pre-trained weights the authors provide in github.

## A.3 Implementation of no-treatment condition

Following the implementation of (Niu et al., 2021), we assume that the model will randomly guess with equal probability for no-treatments. In this case, $Y_{x_R}$, $Y_{x_T}$, and $Y_M$ can be represented as:

$$Y_{x_R} = \begin{cases} y_{x_R} = h_{\theta_{X_R}}(x_R) & \text{if } X_R = x_R \\ y_{x_R^\star} = c & \text{if } X_T = \phi \end{cases} \quad Y_{x_T} = \begin{cases} y_{x_T} = h_{\theta_{X_T}}(x_T) & \text{if } X_T = x_T \\ y_{x_T^\star} = c & \text{if } X_T = \phi \end{cases} \tag{16}$$

$$Y_m = \begin{cases} y_m = h_{\theta_M}(m) & \text{if } X_R = x_R \text{ and } X_T = x_T \\ y_{m^\star} = c & \text{if } X_R = \phi \text{ or } X_T = \phi. \end{cases} \tag{17}$$

We adopt a learnable parameter for c. The uniform distribution assumption for choosing c.

