# OpenReview forum: "Mitigating Uni-modal Sensory Bias in Multimodal Object Detection with Counterfactual Intervention and Causal Mode Multiplexing"
_ICLR.cc/2024/Conference — ICLR 2024 Conference Withdrawn Submission_

### Official Review · Reviewer_AhFd · 2023-10-15

**Soundness:** 3 good
**Presentation:** 2 fair
**Contribution:** 2 fair
**Rating:** 5
**Confidence:** 3

**Summary:**

The paper introduces a framework called Causal Mode Multiplexing (CMM) with the aim at addressing the unimodal sensory bias in multimodal object detection using RGB and thermal sensors. The authors propose this approach based on the usage of counterfactual intervention. CMM leverages two distinct causal graphs for different input data types, enabling effective learning of multimodal reasoning. The authors also present the ROTX-MPed dataset, enriching existing data with some counterexamples of ROTX.

**Strengths:**

A ROTX Multispectral Pedestrian dataset is constructed to include counterexamples which are not present in existing datasets.

Experimental results on various datasets, including the newly introduced ROTX-MPed dataset, showcase the effectiveness of the CMM framework in eliminating sensory bias and enhancing multimodal reasoning in object detection.

**Weaknesses:**

The observation of the performance drop is not significant and the bias from uni-modal to multi-modal with ROTX setting is not a surprising finding.

Compared to the single-modal methods, such as in Table2, the proposed CMM method does not show the advancement. For example, the results of yolo on ROTX-MPed are much better than the RGBT CMM methods.

The description of the new dataset is not sufficient. It is more reasonable to compare other existing datasets.

**Questions:**

How about the statistical information of the new collected dataset? Also, the information of the collection process is less included. Only a few are presented in Appendix A.1. It would be better to compare the new dataset with other existing ones in more detail.

Besides, there are only 1500 pairs of RGB and thermal images. It is less diverse as used for training and testing.

How are the training of the yolo and faster rcnn used in Table 2? Are they trained with only RGB images or RGBT images?

The CMM framework is based on the Uncertainty-guided model with FPN and ResNet50 as backbone. How about the proposed framework on other architectures like vision transformer or MLP-based models?

---

### Official Review · Reviewer_A42c · 2023-10-31

**Soundness:** 2 fair
**Presentation:** 1 poor
**Contribution:** 2 fair
**Rating:** 3
**Confidence:** 4

**Summary:**

This paper proposes a Causal Mode Multiplexing (CMM) framework based on counterfactual intervention to address the issue of uni-modal sensory bias in multimodal learning. The authors introduce different causal graphs for different input types and propose a causal mode multiplexing loss to optimize the exchange between two causal graphs. Additionally, the effectiveness of the proposed method is validated on the created ROTX Multispectral Pedestrian dataset and its generalization is tested on multiple other datasets.

**Strengths:**

1. It is meaningful to address the issue of uni-modal sensory bias in multimodal learning.
2. The creation of the ROTX Multispectral Pedestrian dataset is important for advancing related fields.

**Weaknesses:**

1. Why not create a dataset that includes ROTO, RXTO, and ROTX data types together and train the model using all three types of data to achieve genuine all-weather, all-day detection?
2. Figure 1 seems counterfactual. The purpose of introducing multimodal data is to complement the deficiencies of single-modal data (usually RGB). In Figure 1(b), the RGB data already performs well in object detection. I personally tested several sets of ROTX data from the paper using YOLOx, Faster R-CNN, and DETR, and found no cases of missed detections.
3. How would the results be if the ROTX input type is tested only with RGB data and the RXTO input type is tested only with T data? How does your method improve fusion performance when the input type is ROTX or RXTO? What evidence supports this?
4. To my knowledge, there have been many works studying uni-modal sensory bias in multimodal learning, such as "Balanced Multimodal Learning via On-the-fly Gradient Modulation" (CVPR 2022). Therefore, your contribution is not accurately summarized.
5. Some results in Table 2 are unusual. The AP of YOLO-v5 using only RGB data on the ROTX-MPed dataset is higher than the AP of CMM using RGB+T data on the same dataset. What is the significance of introducing RGBT data?
6. Why not validate the generalization of the proposed method on other tasks based on RGBT data, such as RGBT tracking and RGBT saliency object detection?
7. The writing is not concise and not easily understandable.

**Questions:**

See above weaknesses.

---

### Official Review · Reviewer_6EfW · 2023-11-07

**Soundness:** 2 fair
**Presentation:** 3 good
**Contribution:** 3 good
**Rating:** 6
**Confidence:** 3

**Summary:**

This paper proposes a causality learning approach to the problem of multi-modality object detection where inputs are of different modalities. The presented experimental results show that the proposed method effectively alleviates the uni-modal bias problem identified in multi-modality learning tasks. The paper also contributed a new dataset ROTX-MPed, including counter-examples not available in previous datasets.

**Strengths:**

1. Overall it is a well-written and easy-to-follow paper
2. The proposed new dataset could be a useful contribution to the research community
3. Experimental results have, I believe, cogently illustrated the value of the proposed method and dataset.

**Weaknesses:**

1. A major concern is that ablation experiments, and hence more insightful discussions on authors’ design choices, are missing - can the authors justify why they do not include ablation studies? The experiment results have some vague interpretations. For instance, there are no instructions explaining why Table 2 only displays MR results on KAIST and CVC-14, but not FLIR.
2. Considerations and experiments regarding the time complexity/overhead might be a worthy add to the proposed method.
3. The claims of the proposed dataset require more explanations or experiments to support them. According to the main results, the uni-modal sensory bias problem is solved, especially under nighttime conditions, and the RGBT fusion detection should perform better than uni-modal detection. The results trained on KAIST are satisfactory, and those trained on CVC-14 are also competitive. However, the results trained on FLIR are contrary.
4. Under equation (11), “L” - typo?

**Questions:**

1. It would be helpful to my judgement of its novelty if the authors can justify that their proposed method does not merely involve a direct adoption of causal learning methods but is instead with some insights into the task at hand.
2. Authors are encouraged to explain why ablation studies are not presented.